# An Analysis of Respiratory Muscle Paralysis of Adult Patients in Guillain–Barré Syndrome: A Retrospective Analysis

**DOI:** 10.3390/medicina59071267

**Published:** 2023-07-07

**Authors:** Anqi Wang, Xiaojing Wang, Xinrui Wang, Guozhong Li, Di Zhong

**Affiliations:** 1Department of Neurology, First Affiliated Hospital of Harbin Medical University, Harbin 150000, China; kiki081323@163.com (A.W.); wxjsd_doctor@163.com (X.W.); wxy1986aa@163.com (X.W.); 2Department of Neurology, Heilongjiang Provincial Hospital, Harbin 150000, China; lidoctor097@163.com

**Keywords:** Guillain–Barré syndrome, neuromuscular disease, respiratory muscle paralysis, respiratory failure, mechanical ventilation

## Abstract

Respiratory muscle paralysis is known as a very common complication of Guillain–Barré syndrome (GBS). However, most research has focused on its later stages rather than its earlier stages, including the prognosis of patients with this condition, or factors that act as early predictors of risk. Therefore, our study aimed to identify early predictors of respiratory muscle paralysis in patients with GBS and determine the short-term prognosis of such patients. We recruited 455 GBS patients (age ≥ 18) who had been hospitalized in the First Affiliated Hospital of Harbin Medical University between 2016 and 2021, retrospectively. We recorded clinical and laboratory data and used linear and logistic regression analysis to investigate the relationship between early clinical, examination results, and subsequent respiratory muscle paralysis. Among the 455 patients, 129 were assigned to a respiratory muscle paralysis group and 326 were assigned to a non-respiratory muscle paralysis group. Compared with the non-affected group, the time from onset to admission was shorter (*p* = 0.0003), and the Medical Research Council (MRC) score at admission and discharge was smaller in the affected group (*p* < 0.0001). Compared with the non-affected group, the affected group had higher Hughes and Erasmus GBS Respiratory Insufficiency Score (EGRIS) scores at admission and longer hospital stays (*p* < 0.0001). Patients in the affected group were more likely to have bulbar palsy and lung infections (*p* < 0.0001). To conclude, bulbar palsy, a higher EGRIS score and Hughes score at admission, a lower MRC score, and a shorter time between onset and admission, are all predictive risk factors for respiratory muscle paralysis in patients with GBS. An increase in any of these factors increases the risk of muscle paralysis. Patients with respiratory muscle paralysis have a poorer short-term prognosis than those without respiratory muscle paralysis. Therefore, we should attempt to identify patients with one or more of these characteristics in the early stages of admission, provide ventilation management, and administer IMV treatment if necessary.

## 1. Introduction

Guillain–Barrés syndrome (GBS), also known as demyelinating polyneuritis, is an acute muscle paralysis caused by the immune system damaging the peripheral nervous system [1]. The first symptoms of GBS are numbness, paresthesia, pain, or other symptoms. The typical clinical features are progressive limb weakness and a reduction or disappearance of the tendinous reflex [2]. The onset of GBS can be very rapid and the clinical symptoms usually peak after just 2 weeks. Polyradiculoneuropathy can be detected in nerve conduction studies and cerebrospinal fluid analysis can reveal albumin cytological dissociation; however, both tests can be normal in the early stages of GBS [3,4,5,6,7]. Approximately 25% of patients with GBS may suffer from respiratory insufficiency [4]. The neurological function of most patients can essentially recover within weeks to months, although 20% of patients may have a bad prognosis [4,6]. All patients should be hospitalized, with eligible patients receiving continuous cardiac and respiratory monitoring in the intensive care unit until the disease no longer progresses. Besides, immunotherapy and rehabilitation therapy should be performed as soon as possible [8]. The hospital mortality rate associated with GBS is 3–13% [6,9] and is highest in the convalescence stage (neurologic improvement) [9]. Mortality is predominantly caused by complications such as respiratory failure, infection, hypotension, and severe arrhythmia [6]. Respiratory failure is the most serious short-term complication of GBS and up to 30% of patients require invasive mechanical ventilation (IMV) [10]. However, 60% of patients with intubation may develop complications, including pneumonia, sepsis, and pulmonary embolism. Previous research showed that delayed intubation may increase the risk of pneumonia caused by inhalation and thus worsen the prognosis [11].

Previous studies have suggested that bulbar palsy and facial palsy, the rapid progression of the disease, age, prodromal infection, electrophysiological typing, and severe autonomic nerve dysfunction, are all factors that can influence respiratory muscle paralysis and lead to a poor prognosis in patients with GBS [10,11,12]. In 2010, Walgaar’s team established the Erasmus GBS Respiratory Insufficiency Score (EGRIS) to evaluate the risk of mechanical ventilation in GBS patients within 1 week of admission [11]. Respiratory muscle paralysis is known as a very common short-term complication of GBS; however, since the early symptoms of GBS patients with respiratory muscle paralysis are mild, most studies focus on patients requiring mechanical ventilation at later stages, including the prognosis of patients with this condition, or factors that act as early predictors of risk. Therefore, in the present study, we attempted to identify early predictors of respiratory muscle paralysis in patients with GBS and determine the short-term prognosis of such patients.

## 2. Materials and Methods

### 2.1. Patient Selection 

We recruited patients who had been diagnosed with GBS according to the standard diagnostic criteria for Guillain–Barré syndrome [6] and were hospitalized in the First Affiliated Hospital of Harbin Medical University between 2016 and 2021, retrospectively. 

The inclusion criteria were as follows: (1) the patient was over 18 years of age, (2) the patient had not received intravenous immunoglobulin (IVIG) or plasma exchange prior to admission, and (3) the patient provided a signed informed consent form. 

The exclusion criteria were as follows: (1) respiratory muscle paralysis was caused by another form of peripheral neuropathy, (2) diagnosis had excluded each type of GBS classification, and (3) there was an incomplete set of clinical data. 

### 2.2. Grouping Criteria

Patients were divided into a respiratory muscle paralysis group (the affected group) or a non-respiratory muscle paralysis group (the control group) based on whether one or more of the following conditions were met or not, respectively: (1) breathlessness that was characteristically worse when bending forward (e.g., when tying shoelaces or getting out of a car) or lying flat, or could not complete full sentences with a single breath [13,14]; (2) typical signs of neuromuscular respiratory failure including paradoxical abdominal motion (abdomen moved inwards instead of outwards during inhalation) [15]; (3) respiratory failure (a PaO_2_ ≤ 60 mmHg without oxygen or an oxygenation index < 300 with oxygen), or (4) mechanical ventilation (MV). 

According to these grouping criteria, the affected group was further divided into three sub-groups according to the severity of respiratory muscle paralysis. Group 1 (the slightly affected group) only had symptoms of respiratory muscle paralysis; no significant reduction was observed in blood gas analysis or blood gas analysis revealed a PaO_2_ > 60 mmHg or an oxygenation index > 300% without oxygen inhalation. Group 2 (the respiratory failure group) featured patients with respiratory failure (a PaO_2_ ≤ 60 mmHg without oxygen or an oxygenation index < 300 with oxygen), but without IMV (no endotracheal intubation/tracheotomy). Group 3 (the MV group) featured patients with IMV (with endotracheal intubation/tracheotomy). 

### 2.3. Data Collection

We collected a range of data for each patient on enrollment, including patient age, gender, the duration from admission to onset, infections preceding GBS, the total length of hospitalization, bulbar palsy, facial palsy, and electrophysiological type (tested on admission, and conforming to Hadden criteria [16]). We also collated the EGRIS score and Hughes score at admission, and the Medical Research Council (MRC) score [17] at admission and discharge. The MRC sum score was defined as the sum of MRC scores from six different muscles, including shoulder abduction, forearm flexion, wrist extension, thigh flexion, knee extension and foot dorsiflexion (measured bilaterally); the resultant scores ranged from 0 (tetraplegic) to 60 (normal).

EGRIS scores were shown in Table 1; an EGRIS score of 0–2 was classified as low risk, a score of 3–4 was classified as moderate risk, and a score of 5–7 was classified as high risk. The Hughes score was a widely accepted scale used to assess the functional status of GBS patients and ranged from 0 (normal) to 6 (death). Hughes score < 3 was classified as mild while a Hughes score ≥ 3 was classified as severe. 

### 2.4. Statistical Analysis

SPSS software was used for statistical processing, and GraphPad Prism was used to draw Forest plots. Data were expressed as a mean ± standard deviation (normal distribution) or as quartiles (skewed distribution). The t-test or rank sum analysis was used to compare data between the two groups. Analysis of variance or multiple rank sum analysis combined with multiple analysis was used to compare the data between multiple groups. The Chi-squared test or Fisher’s exact test was used to compare numerical data. Linear or logistic regression analysis was used for correlation analysis, and 95% confidence intervals (CIs) were expressed as Forest plots. *p* < 0.05 was considered statistically significant.

## 3. Results

### 3.1. Basic Clinical Data 

In total, we recruited 455 subjects, including 129 in the affected group and 326 in the control group. There were 284 males and 171 females. The mean age was 51.8 ± 16.3 years. In total, 382 patients (84.0%) received immunotherapy (IVIG and/or plasmapheresis), 374 (82.2%) received only IVIG, 1 (0.2%) received only plasmapheresis, and 7 (1.6%) received both IVIG and plasmapheresis. There were 107 (23.5%) patients with facial nerve palsy, 129 (28.4%) with bulbar palsy, and 110 (24.4%) with pulmonary infection. Analysis showed that 216 patients (47.5%) had infections preceding GBS, including 116 (25.5%) who had a history of upper respiratory infection (URI), 64 (14.1%) who had diarrhea, 11 (2.4%) had both URI and diarrhea, and 25 (5.5%) had a history of another type of infection including vaccination, other intestinal diseases, and surgical history. There were no significant differences in gender, age, therapy, facial paralysis and infection between groups (*p* > 0.05, Table 2).

### 3.2. Comparisons between the Affected Group and the Control Group

Detailed statistical results for the two groups were shown in Table 2 while the correlation analysis results and Forest plots were shown in Figure 1 and Figure 2, respectively, including linear regression and logistic regression analysis. Rank sum analysis showed that there were statistically significant differences between the groups with regards to the time from onset to admission (*p* < 0.001), the MRC score on admission (*p* < 0.001), MRC score on discharge (*p* < 0.001), Hughes score on admission (*p* < 0.001), EGRIS score on admission (*p* < 0.001), and the total length of hospitalization (*p* = 0.002). The Chi-squared test showed that there were statistically significant differences between the groups with regard to electromyographic typing, and the proportion of bulbar palsy and pulmonary infection (*p* = 0.013). Multiple comparison analysis of electromyographic typing showed that there were statistically significant differences between the two groups in terms of pure demyelination type and both types (*p* < 0.05), but there was no statistical difference between the two groups with regards to the same type of EMG classification (*p* > 0.05). Linear regression analysis showed that the affected group had a shorter onset to admission (*p* = 0.0003), a smaller MRC score on admission (*p* < 0.0001), a smaller MRC score at discharge (*p* < 0.0001), a larger Hughes score (*p* < 0.0001) and EGRIS score (*p* < 0.0001), and a longer total hospital stay (*p* < 0.0001) than the control group. Logistic regression analysis showed that compared with the control group, the odds ratios (ORs) for bulbar palsy and pulmonary infection in the affected group were both greater than 1 (*p* < 0.0001), thus indicating that the patients in the affected group were likely to have bulbar palsy and pulmonary infection.

### 3.3. Intra-Group Comparative Analysis of the Affected Groups

Comparisons of the general data and clinical data for the three groups were shown in Table 3 while correlation analysis results and associated Forest plots were shown in Figure 3 and Figure 4, respectively, including linear regression and logistic regression analysis. There were significant differences in age, time from onset to admission, MRC score at admission, MRC score at discharge, Hughes score at admission, and EGRIS score at admission between the two groups (*p* < 0.05). The Chi-squared test showed that the overall difference was statistically significant with regard to the proportions of pulmonary infection in each group (*p* < 0.001). Linear regression analysis showed that patients in group 2 and group 3 were older than those in group 1 (*p* = 0.0482 and *p* = 0.0242), and the total length of hospital stay was longer (*p* = 0.0235 and *p* = 0.0019). Patients in group 3 had a lower MRC score on admission and a lower MRC score at discharge (*p* < 0.0001); they also had a higher Hughes score and EGRIS score on admission when compared with patients in group 1 (*p* < 0.0001 and *p* = 0.0001). Logistic regression analysis showed that compared with those in group 1 the OR values of pulmonary infection for patients in group 2 and group 3 were all greater than 1 (*p* = 0.0033), thus indicating that patients in group 2 and group 3 were more likely to have pulmonary infection than those in group 1.

## 4. Discussion

Our research aimed to investigate the risk factors and their correlation with respiratory muscle paralysis in patients with GBS and the short-term prognosis of patients with respiratory muscle paralysis. Combined with previous studies, the results of our research demonstrated that bulbar palsy and facial palsy, the time from onset to admission, different types of prodromal infection, electrophysiological typing, and Hughes, EGRIS and MRC scores on admission, may be the most significant factors influencing respiratory muscle paralysis and a poor prognosis in GBS patients.

Previous studies have suggested that bulbar palsy increases the risk of MV and is more likely to develop into severe GBS [7,18]. Intergroup comparisons between the affected group and the control group revealed the proportion of patients with bulbar palsy in the affected group was 36.2% higher than that in the control group. Consistently, respiratory and bulbar paralysis are common characteristics of GBS, bulbar palsy can lead to the abnormal contraction of the throat muscles, which results in dysphagia, bucking, and other symptoms. Therefore, the ability to clear respiratory secretions is reduced, thus causing lung infection and aggravation of the condition, as demonstrated in our current research [19].

Studies carried out in Japan found that the EGRIS score has a good predictive value for whether GBS patients will develop respiratory muscle paralysis one week after admission [20]. In the present study, we found that the EGRIS score of the affected group was higher than that of the control group, which was confirmed by linear regression analysis. In line with our results, it has been proved that GBS patients with EGRIS scores < 5 had better functional recovery compared to those with score ≥ 5 [21]. Comparing the data distribution histograms and box graphs for the two groups (Figure 5), we found that data from the control group were mostly low-risk and low-medium risk while data from the affected group followed an approximately normal distribution, dominated by medium-high risk cases. Intra-group comparison of the affected group revealed that data from the non-mechanical ventilation group were mostly from low-medium risk areas while data from the MV group were mostly from medium-high risk, and the difference was statistically significant. These data suggest that the EGRIS score can also be used to predict whether patients will develop respiratory muscle paralysis or not, which was consistent with previous studies [22,23]. The high-risk classification indicates that the patient is highly likely to need MV at an advanced stage. However, if the patient is classified as low risk, it is still possible to develop respiratory-respiratory paralysis. Therefore, we suggest that respiratory ventilation should be managed and monitored even if a patient’s EGRIS score is classified as low risk. Previous studies have shown that intermediate- and high-risk patients, particularly those with severe globular paralysis, should be selectively intubated at an early stage, based on the EGRIS score, as this is a better option than noninvasive ventilation (NIV) [15,24].

The Hughes score is used as a disability score for GBS and can be used to classify patients with GBS into mild and severe groups. A higher Hughes score on admission indicates a lower muscle strength score and possible respiratory involvement. We also found that the MRC score, as an indicator of muscle strength, had a similar clinical value as the Hughes score. Our inter-group and intra-group comparisons showed that the lower the MRC score, the higher the possibility and severity of respiratory muscle involvement in GBS patients, which was consistent with previous studies [25]. 

The nerves that innervate the respiratory muscles are anatomically adjacent to those that innervate movement in the upper limbs. Based on this information, we hypothesized that the extent of injury in the muscle strength of the upper limbs would be more severe than that in the lower limbs in the affected group. At the same time, to avoid the influence of short onset time on our results, we determined the total muscle strength in the upper and lower limbs during the peak period (the worst muscle strength) of patients in the affected group. However, in contrast to our expectations, we found that the muscle strength in the lower limbs was significantly worse than that in the upper limbs. Whether this result is related to the pathogenesis of GBS or the fact that most patients in this study were patients with heavier lower limbs need to be further clarified in a follow-up study. 

Although there was no significant difference in overall electromyographic typing in the affected group, 48.1% of patients with axonal injury alone were detected in the MV group, while only 3.7% of patients with demyelination alone were detected. In one previous study, 46% of demyelinating patients required mechanical ventilation at a later stage, while only 17% of patients in the ambiguous group required mechanical ventilation, and no patients in the axonal injury group required mechanical ventilation [26]. These results contradict our present results. This phenomenon may be related to the different proportions of GBS subtypes in different regions [27]. 

In the present study, we estimated short-term prognosis by simply comparing the two groups of patients with regards to the duration of hospitalization, MRC scores ≥ 48 points (meaning two or more limb muscle strength grades ≥ 4 points and that the patient could stand) and MRC scores ≤ 36 points (meaning two or more limb muscle strength grades ≤ 3 points and that the patient could not stand up). We found that patients in the affected group were hospitalized for longer than those in the control group. Intra-group comparisons also showed that patients on MV had longer hospital stays than those with a mild presentation. Likewise, it has been reported that the requirement for MV was significantly associated with the short-term outcome of severe GBS [28]. There were a few patients in the affected group that abandoned treatment halfway through. Of the 455 patients enrolled, three patients died during hospitalization; all of these were in the affected group and the MV group, thus accounting for 2.3% and 11.1% of the total number of patients in the affected group and MV group, respectively; the mortality rate in such cases was previously reported to be 10–20% [29,30]. This may be associated with more severe disease and peripheral nerve injury in patients with respiratory muscle paralysis/MV, thus resulting in a more difficult recovery in both types of patients than in patients without muscle paralysis/MV or those with only mild involvement. We found that more than 50% of patients in the affected group may still be in bed or unable to take care of themselves after discharge, while more than 75% of the patients in the control group could take care of themselves. In summary, patients in the respiratory muscle paralysis group had a poorer prognosis at an earlier stage than patients in the control group. In addition, it has been reported that the long-term prognosis of GBS patients using MV suggested that the long-term quality of life was impaired in a considerable number of patients, including chronic pain, a reduced ability to perform pre-disease activities, and decreased mobility [31].

There are also limitations in our study. First, our study is a retrospective analysis and lacks follow-up observation to analyze the long-term prognosis of GBS. Second, the sample size of this study is insufficient for stratified analysis, and further prospective studies are needed to confirm our findings. Finally, because our study is based in a single center, the living environment of the recruited patients was similar, which may affect our results. Therefore, further multi-center studies are now required to verify the results of this study.

## 5. Conclusions

The presence of bulbar palsy, a higher EGRIS score, a Hughes score at admission, a lower MRC score, and a shorter time between onset and admission were found to be predictive factors of respiratory muscle paralysis in GBS patients. An increase in any of these factors increases the likelihood of MV in the later stages of the disease. The short-term prognosis of patients with respiratory muscle paralysis is worse than that of patients without such paralysis. Therefore, we should attempt to identify patients with one or more of these characteristics in the early stages of admission, provide ventilation management, and administer IMV treatment if necessary. 

## Figures and Tables

**Figure 1 medicina-59-01267-f001:**
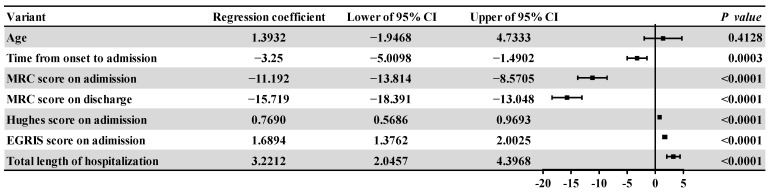
Forest plot of the regression coefficients and 95% confidence intervals of the affected group relative to the control group. Note: CI: confidence interval. MRC: Medical Research Council. EGRIS: Erasmus GBS Respiratory Insufficiency Score.

**Figure 2 medicina-59-01267-f002:**
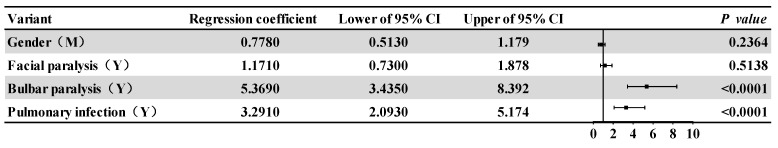
Forest plots of the OR values and 95% confidence intervals are in the affected group relative to the control group. Note: CI: confidence interval.

**Figure 3 medicina-59-01267-f003:**
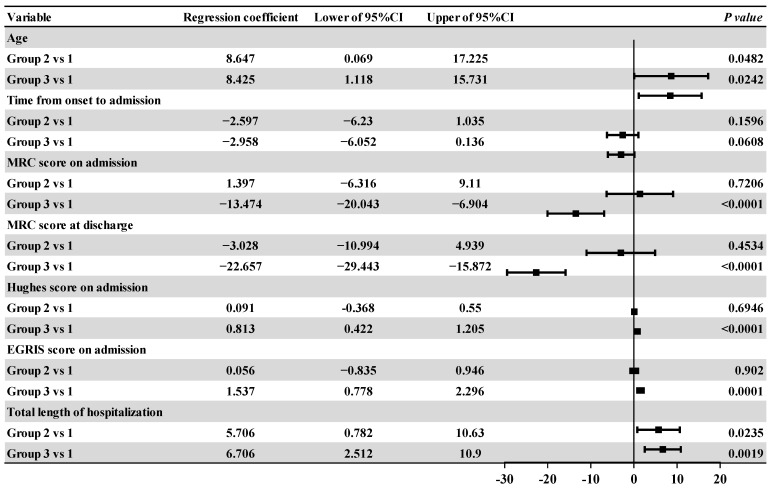
Forest plot of the regression coefficients and 95% confidence intervals for patients with different grades of respiratory paralysis relative to group 1. Note: MRC: Medical Research Council. EGRIS: Erasmus GBS Respiratory Insufficiency Score. URI: Upper respiratory infection. CI: confidence interval.

**Figure 4 medicina-59-01267-f004:**
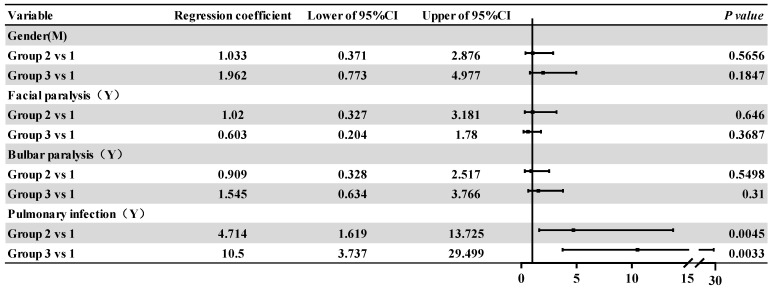
Forest plot of logistic and 95% confidence intervals for different grades of respiratory paralysis relative to group 1. Note: CI: confidence interval.

**Figure 5 medicina-59-01267-f005:**
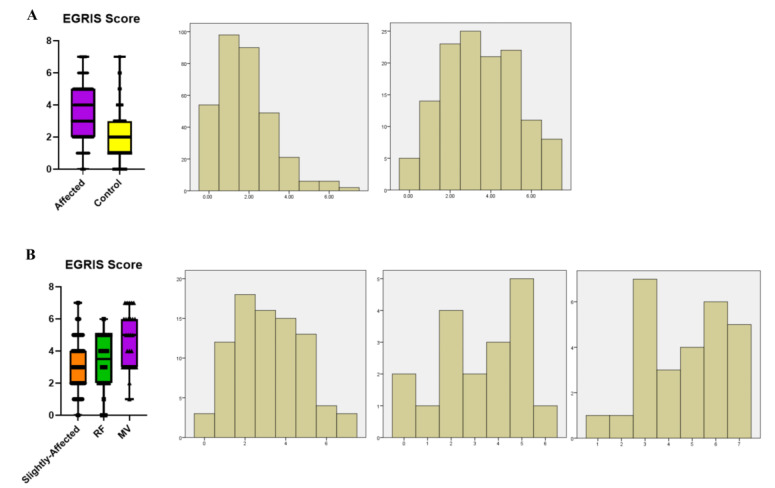
Distribution of EGRIS score. (**A**). Boxplots and histograms of the distribution of EGRIS score data of the affected and control groups. Left: Control group, right: Affected group. The ordinate of the histogram is frequency. (**B**). Boxplots and histograms of the distribution of EGRIS score data within the affected groups. Left: Group l, Middle: Group 2, Right: Group 3. Note: EGRIS: Erasmus GBS Respiratory Insufficiency Score. RF: Respiratory failure. MV: Mechanical ventilation.

**Table 1 medicina-59-01267-t001:** EGRIS score.

Measure	Categories	Score
Time from onset to admission	>7days	0
	4–7 days	1
	≤3 days	2
Facial or bulbar palsy on admission	Absence	0
	Presence	1
MRC score on admission	60–51	0
	50–41	1
	40–31	2
	30–21	3
	≤20	4
Total score		0–7

Note: MRC: Medical Research Council.

**Table 2 medicina-59-01267-t002:** Comparison of clinical features between the affected group and the control group [n (%)].

Item		Affected (n = 129)	Non-Affected (n = 326)	*X*^2^/*t*/*Z*	*p*
Gender					
	Male	75 (58.1)	209 (64.1)	1.405	0.236
	Female	54 (41.9)	117 (35.9)		
Age ^A^		55 (40.5, 66)	53 (40, 62)	−0.948	0.343
EMG type					
Demyelinating		9 (7.0)	51 (15.6)		
Axon		44 (34.1)	113 (34.7)	10.715	0.013
Both have		71 (55.0)	137 (42.0)		
Both have not		5 (3.9)	25 (7.7)		
Time from onset to admission		4 (2, 7)	7 (4, 12)	−5.132	<0.001
Facial paralysis		33 (25.6)	74 (22.7)	0.427	0.514
bulbar palsy		70 (54.3)	59 (18.1)	59.511	<0.001
MRC score ^A^					
	Admission	39 (28, 52)	52.5 (45, 59)	−7.228	<0.001
	Discharge	36 (24, 51)	54 (48, 60)	−9.209	<0.001
Hughes score on admission ^A^		4 (3, 4)	3 (2, 4)	−7.106	<0.001
EGRIS score on admission ^A^		3 (2.00, 5.00)	2 (1, 3)	−8.982	<0.001
Total length of hospitalization ^A^		10 (6, 12.5)	8 (6.75, 10)	−5.132	0.002
Pulmonary infection		53 (41.1)	57 (17.5)	28.084	<0.001
Infection					
URI		36 (27.9)	80 (24.5)	7.010	0.200
Diarrhea		18 (14.0)	46 (14.1)		
URI and diarrhea		5 (3.9)	6 (1.8)		
Other		11 (8.5)	14 (4.3)		
Non		59 (45.7)	180 (55.2)		

The normaly distributed data were expressed in mean ± standard deviation, and we used quartiles to express skewed distributed data. Standard ^A^ data are non-normally distributed and are represented by the quartile method. Note: EMG: Electromyography. MRC: Medical Research Council. EGRIS: Erasmus GBS Respiratory Insufficiency Score. URI: Upper respiratory infection.

**Table 3 medicina-59-01267-t003:** Comparison of clinical features in different severity of affected group [n (%)].

		Patients with Respiratory Muscle Paralysis (n = 129)				
Items		Group 1 (n = 84)	Group 2 (n = 18)	Group 3 (n = 27)	*X*^2^/*F*/*Z*	*p* (α = 0.05)
Gender						
	Male	46 (54.8)	10 (55.6)	19 (70.4)	2.103	0.350
	Female	38 (45.2)	8 (44.4)	8 (29.6)		
Age		49.80 ± 17.103	58.44 ± 16.992	58.22 ± 15.067	3.814	0.025 ^a^
EMG type						
Demyelinating		8 (9.5)	0 (0.00)	1 (3.7)	7.787	0.194
Axon		28 (33.3)	3 (16.7)	13 (48.1)		
Both have		44 (52.4)	14 (77.8)	13 (48.1)		
Both have not		4 (4.8)	1 (5.6)	0 (0.0)		
Time from onset to admission		5 (3,8)	3 (2, 5)	3 (2, 5)	9.426	0.009 ^b^
Facial paralysis		23 (27.4)	5 (27.8)	5 (18.5)	0.896	0.639
bulbar palsy		44 (52.4)	9 (50.0)	17 (63.0)	1.075	0.584
MRC score						
Admission		41.5 (30, 54)	44 (34, 54)	28 (12, 42)	12.636	0.002 ^c^
Discharge		44.5 (30, 54)	37.56 ± 14.513	12 (0, 30)	27.227	<0.001 ^d^
Hughes score on admission		4 (2, 4)	4 (3, 4)	4 (4, 5)	15.021	0.001 ^e^
EGRIS score on admission		3 (2, 4)	3.22 ± 1.833	5 (3, 6)	13.675	0.001 ^f^
Total length of hospitalization		9 (6, 11)	10.5 (7,14)	12 (5, 23)	4.151	0.125
Pulmonary infection		21 (25.0)	11 (61.1)	21 (77.8)	26.979	<0.001
Infection						
URI		25 (29.8)	4 (22.2)	7 (25.9)	6.970	0.505
Diarrhea		11 (13.1)	2 (11.1)	5 (18.5)		
URI and diarrhea		5 (6.0)	0 (0.0)	0 (0.0)		
Other		4 (4.8)	3 (16.7)	4 (14.8)		
Non		39 (46.4)	9 (50.0)	11 (40.7)		

^a^. Group 1 vs. group 2: *p* = 0.145, group 1 vs. group 3: *p* = 0.073, group 2 vs. group 3: *p* = 1.000. ^b^. Group 1 vs. group 2: *p* = 0.164, group 1 vs. group 3: *p* = 0.018, group 2 vs. group 3: *p* = 1.000. ^c^. Group 1 vs. group 2: *p* = 1.000, group 1 vs. group 3: *p* = 0.002, group 2 vs. group 3: *p* = 0.018. ^d^. Group 1 vs. group 2: *p* = 1.000, group 1 vs. group 3: *p* < 0.001, group 2 vs. group 3: *p* = 0.004. ^e^. Group 1 vs. group 2: *p* = 1.000, group 1 vs. group 3: *p* < 0.001, group 2 vs. group 3: *p* = 0.022. ^f^. Group 1 vs. group 2: *p* = 1.000, group 1 vs. group 3: *p* < 0.001, group 2 vs. group 3: *p* = 0.47. Data conforming to normal distribution were represented by mean ± standard deviation, and data not conforming to a normal distribution by the quartile method. Note: EMG: Electromyography. MRC: Medical Research Council. EGRIS: Erasmus GBS Respiratory Insufficiency Score. URI: Upper respiratory infection.

## Data Availability

All data generated during this study are included in this published article and can be provided as requested.

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
