# Peer review of "An Analysis of Respiratory Muscle Paralysis of Adult Patients in Guillain–Barré Syndrome: A Retrospective Analysis"

_medicina, 2023, doi:10.3390/medicina59071267_

Round 1

Reviewer 1 Report

This article analyzed the respiratory muscle paralysis of adult patients in Guillain-Barré syndrome. The whole manuscript is quite clear and interesting. But lots of issues in this study need to be addressed.
1.    There are some grammar errors in your manuscript, please carefully check and revise them. For example: “Age, time from onset to admission, MRC score on admission, MRC score at discharge, Hughes score on admission, and EGRIS score on admission, were significantly different when compared between groups (P<0.05).” should be revised to “There were significant differences in age, time from onset to admission, MRC score at admission, MRC score at discharge, Hughes score at admission and EGRIS score at admission between the two groups (P<0.05).”
2.    Please polish the language of your manuscript with a professional help.
3.    The limitations of your study should be indicated at the end of Discussion section.
4.    Since you have provided the full name of GBS in Abstract section, so the full name of abbreviations such as MRC and EGRIS should be also indicated.
5.    Please add a space between the word and the punctuation mark in your manuscript. For example: “PaO2≤60mmHg” should be revised to “PaO2 ≤60 mmHg.”
6.    In results section, the comparison results of Basic clinical data between the affected group and the control group should be described.

Please see comment 1 and 2

Author Response

This article analyzed the respiratory muscle paralysis of adult patients in Guillain-Barré syndrome. The whole manuscript is quite clear and interesting. But lots of issues in this study need to be addressed.

  1.    There are some grammar errors in your manuscript, please carefully check and revise them. For example: “Age, time from onset to admission, MRC score on admission, MRC score at discharge, Hughes score on admission, and EGRIS score on admission, were significantly different when compared between groups (P<0.05).” should be revised to “There were significant differences in age, time from onset to admission, MRC score at admission, MRC score at discharge, Hughes score at admission and EGRIS score at admission between the two groups (P<0.05).”

Response: Thank you for your suggestion, we have carefully checked and revised the grammar errors in our manuscript.

  1.    Please polish the language of your manuscript with a professional help.

Response: Thank you for your suggestion, we have polished the language of our manuscript with a professional help.

  1.    The limitations of your study should be indicated at the end of Discussion section.

Response: Thank you for your suggestion, we have added the limitations of our study at the end of Discussion section.

  1.    Since you have provided the full name of GBS in Abstract section, so the full name of abbreviations such as MRC and EGRIS should be also indicated.

Response: Thank you for your suggestion, we have provided the full name of MRC and EGRIS in Abstract section.

  1.    Please add a space between the word and the punctuation mark in your manuscript. For example: “PaO2≤60mmHg” should be revised to “PaO2 ≤60 mmHg.”

Response: Thank you for your suggestion, we have added a space between the word and the punctuation mark in our manuscript.

  1.    In results section, the comparison results of Basic clinical data between the affected group and the control group should be described.

Response: Thank you for your suggestion, we have indicated the comparison results of Basic clinical data between the affected group and the control group in Results section.

Reviewer 2 Report

Dear authors, Revise your paper according to the comments provided to you by the reviewers. 

In Abstract section, please add P values in the description of results.

What are the clinical manifestations of Guillain-Barrés syndrome? Is there anything special about it compared to other neuropathies? A brief introduction to them is recommended.

The discussion section should not be a repetition of the results, it is suggested to compare your results with previous studies to indicate the highlights of your study.

“Hughes score is a widely accepted scale that us used to assess the functional status of GBS patients and ranges from 0 (normal) to 6 (death).” Please delete “us.”

The abbreviations in the tables and figures should be explained.

The conclusion part is the summary of the article, which should briefly and clearly describe the highlights and research significance of your study.

NA

Author Response

Dear authors, Revise your paper according to the comments provided to you by the reviewers.

In Abstract section, please add P values in the description of results.

Response: Thanks for your suggestion, we have added P values in the description of results.

What are the clinical manifestations of Guillain-Barrés syndrome? Is there anything special about it compared to other neuropathies? A brief introduction to them is recommended.

Response: Thanks for your suggestion, we have added the clinical manifestations of Guillain-Barrés syndrome in Introduction section.

The discussion section should not be a repetition of the results, it is suggested to compare your results with previous studies to indicate the highlights of your study.

Response: Thank you for your suggestion, we have compared our results with previous studies in Discussion section.

“Hughes score is a widely accepted scale that us used to assess the functional status of GBS patients and ranges from 0 (normal) to 6 (death).” Please delete “us.”

Response: Thank you for your suggestion, we have deleted “us” in this sentence.

The abbreviations in the tables and figures should be explained.

Response: Thank you for your suggestion, we have provided the full name of abbreviations in the tables and figures.

The conclusion part is the summary of the article, which should briefly and clearly describe the highlights and research significance of your study.

Response: Thank you for your suggestion, we have revised the conclusion part to be brief and clear.

Reviewer 3 Report

This paper is about an analysis of respiratory muscle paralysis of adult patients in Guillain-Barré syndrome. The current script can be revised and added with clarifications of some equivocal expressions to achieve publication quality. The comments are as follows:

·        The aim of your study should be clearly indicated in Abstract section, in order to make readers to understand the main idea of your study.

·        A forward-looking conclusion should be added at the end of Abstract section to highlight the novelty and clinical significance of your study.

·        In Introduction section, the current main treatments for Guillain-Barrés syndrome should be briefly described, so that readers can learn more about this disease.

·        The title of your study was “An analysis of respiratory muscle paralysis of adult patients in Guillain-Barré syndrome,” but the inclusion criteria was “the patient was over 16 years-of-age,” The come of age for an adult should be over 18, so it is suggested to revise to “the patient was over 18 years-of-age.”

·        The description of results should be provided with more details. For example, the p values of the results of Logistic regression analysis were not indicated.

·        In Results section, which table provides the Basic clinical data of patients? Please indicate it in the section of Basic clinical data.

·        There are many incorrect tenses in the description of this article. For example, in the sentence of “Detailed statistical results for the two groups are shown in Table 2 while the correlation analysis results and Forest plots are shown in Figure 1 and Figure 2, respectively, including liner regression and logistic regression analysis,” it should revise “are” into “were.”

Moderate improvement in english language is needed.

Author Response

This paper is about an analysis of respiratory muscle paralysis of adult patients in Guillain-Barré syndrome. The current script can be revised and added with clarifications of some equivocal expressions to achieve publication quality. The comments are as follows:

  •        The aim of your study should be clearly indicated in Abstract section, in order to make readers to understand the main idea of your study.

Response: Thanks a lot for your kind advice, we have added the aim of our study in Abstract section.

  •        A forward-looking conclusion should be added at the end of Abstract section to highlight the novelty and clinical significance of your study.

Response: Thanks for your advice, we have add a forward-looking conclusion at the end of Abstract section to highlight the novelty and clinical significance of your study.

  •        In Introduction section, the current main treatments for Guillain-Barrés syndrome should be briefly described, so that readers can learn more about this disease.

Response: Thank you for your suggestion, we have added the current main treatments for Guillain-Barrés syndrome in Introduction section.

  •        The title of your study was “An analysis of respiratory muscle paralysis of adult patients in Guillain-Barré syndrome,” but the inclusion criteria was “the patient was over 16 years-of-age,” The come of age for an adult should be over 18, so it is suggested to revise to “the patient was over 18 years-of-age.”

Response: Thank you for your suggestion, we have revised “16” into “18” in the inclusion criteria.

  •        The description of results should be provided with more details. For example, the p values of the results of Logistic regression analysis were not indicated.

Response: Thank you for your suggestion, we have described the results with more details.

  •        In Results section, which table provides the Basic clinical data of patients? Please indicate it in the section of Basic clinical data.

Response: Thank you for your suggestion, we have indicated that Table 2 provided the Basic clinical data of patients.

  •        There are many incorrect tenses in the description of this article. For example, in the sentence of “Detailed statistical results for the two groups are shown in Table 2 while the correlation analysis results and Forest plots are shown in Figure 1 and Figure 2, respectively, including liner regression and logistic regression analysis,” it should revise “are” into “were.”

Response: Thanks for your suggestion, we have revised the incorrect tenses in our manuscript.

Round 2

Reviewer 1 Report

The authors have satisfactorily resolved all of my concerns. 

Reviewer 3 Report

Accepted. No more changes are required.